# Implementation and Maintenance of a Community-Based Intervention for Refugee Youth Reporting Symptoms of Post-Traumatic Stress: Lessons from Successful Sites

**DOI:** 10.3390/ijerph18010043

**Published:** 2020-12-23

**Authors:** Elin Lampa, Anna Sarkadi, Georgina Warner

**Affiliations:** Department of Public Health and Caring Sciences, Uppsala University, 751 23 Uppsala, Sweden; anna.sarkadi@pubcare.uu.se (A.S.); georgina.warner@pubcare.uu.se (G.W.)

**Keywords:** trauma, child, refugee, mental health, implementation, community-based intervention

## Abstract

Over the last few years there have been attempts to scale-up Teaching Recovery Techniques (TRT), a community-based group intervention for refugee youth reporting symptoms of post-traumatic stress, across Sweden using the distribution network pathway model. This implementation model allows for quick spread, but only for a low level of control at local sites. This study explores factors and agents that have facilitated the implementation and maintenance of the community-based intervention in successful sites. Seven semi-structured interviews were conducted with personnel from “successful” community sites, defined as having conducted at least two groups and maintaining full delivery. Data were analyzed using content analysis to identify a theme and categories. The main theme “Active networking and collaboration” was key to successful maintenance of community-based delivery. Categories included “Going to where the potential recipients are”, relating to the importance of networks, and “Resource availability and management for maintenance”, relating to the challenges due to the lack of a lead organization supplying necessary funds and support for maintenance. Additionally, “Careful integration of the interpreter” underlined that interpreters were essential co-facilitators of the intervention. Although the interviewed professionals represented successful sites, they remained dependent on informal networks and collaboration for successful maintenance of community-based delivery.

## 1. Introduction

Transferring effective health and wellbeing interventions into community-based settings is challenging, with many only ever implemented in the academic settings in which they were developed [1]. The literature on implementation frameworks, summarized in a review article by Meyers, Durlak and Wandersman [2], suggests that factors relating to the innovation per se, the implementing organizations and individual professionals as well as the implementation support strategies employed, may influence implementation. The authors suggest initial considerations, structure and clear responsibilities throughout the implementation process, and learning from experiences is key to successful implementation [2].

Replicating effective programs (REP) is an implementation framework that offers strategies to maximize the chances for sustaining an intervention [3]. The framework proposes that factors contributing to successful implementation are present at every stage of the process. This includes pre-conditions such as the presenting need and existing services within a community, pre-implementation factors including logistics planning and staff training, implementation requirements such as ongoing support, maintenance requirements such as incorporation into job duties, and evolution requirements such as intervention refinement to address shifting needs profiles. There can be variation in the particular factors instrumental in implementation success depending on the attributes of the intervention and the context in which it is being implemented. It is important to take lessons from community sites where interventions appear to have been implemented successfully in order to further enhance knowledge for future implementation efforts. In a recent realist review on community-based mental health services for refugees, very diverse factors affecting delivery were identified, suggesting that the complex real-life implementation of such services needs to be flexible to the unstable circumstances of the recipients, including task-shifting for delivery closer to the recipients [4].

### 1.1. Setting: Refugee Minors in Sweden

In 2015, over 70,000 refugee minors arrived in Sweden [5]. Symptoms of post-traumatic stress are well-known sequels of war and forced migration due to risk of human rights violations and persecution, and there have been calls for the urgent need to put this issue on the international agenda [6]. Refugee minors have full access to healthcare in Sweden, but the sheer number made it impossible for the existing healthcare and welfare systems to effectively address their mental health needs. Today, many of these refugees, especially those who arrived as unaccompanied minors, are young adults. Those having been granted asylum or a residence permit have full access to healthcare, but asylum seekers and undocumented migrants have limited access to healthcare. Despite formal access to healthcare, however, issues related to stigma, mental health literacy, and lack of knowledge in how to navigate the healthcare system still pose barriers to receiving psychoeducation and effective treatment.

### 1.2. Teaching Recovery Techniques: A Community-Based Group Intervention for Refugee Youth

Teaching Recovery Techniques (TRT) is a manualized group intervention based on trauma-focused cognitive behavioural therapy (TF-CBT) developed by the Children and War Foundation in Norway and the United Kingdom [7,8]. The intervention aims to increase coping and promote recovery from post-traumatic stress disorder (PTSD) in children aged eight or above in conflict or disaster and was explicitly built for use in low-resource settings, where large numbers of children needed intervention. Each TRT group is facilitated by two group leaders, who have received a three-day training. The intervention is delivered mainly in the community, by group leaders who do not need to have therapeutic training, but experience of working with children or youth. Therefore, it goes well in line with the increased demand on community-based, accessible mental health support, especially for refugees [9]. Safety aspects are considered and if a participant in a TRT group presents with profound symptoms of mental health problems they will be referred to specialist care based on a set safety protocol.

The five weekly group sessions for the children focus on psychoeducation and strategies to reduce trauma symptoms. The techniques are expected to be delivered in the order they appear in the manual, however the manual leaves room for some flexibility in the delivery (see Table 1). In Sweden, the training also encourages including an initial “getting to know each other session” and a “follow up session” after the five sessions. Additionally, the programme includes two sessions for the children’s caregivers, which focus on psychoeducation and how they can support the child.

TRT has had promising impact on clinical outcomes. Trials of TRT in Gaza [10,11], during ongoing conflict, reported decreases in symptoms of both PTSD and depression. A trial in Thailand after the tsunami [12] reported decreases in PTSD symptoms. Similarly, an exploratory trial of TRT with unaccompanied refugee youth in Sweden showed decreased symptoms of PTSD and depression [13]. In the same study, a qualitative evaluation showed high acceptability of the programme, with the recipients expressing that psychoeducation and the group format reduced stigma and provided relief in that posttraumatic stress is a normal reaction to extreme events [13]. The TRT manual is based on TF-CBT, which has a strong evidence base as individual therapy and is known to be an effective treatment for PTSD [14,15,16]. However, similar to TRT and most psychological interventions, it has not been extensively tested in different cultural settings [14]. In comparison, TRT is a more light-touch intervention aiming to reach large numbers of children. There are two ongoing randomized controlled trials of TRT in Sweden [17,18]. The present study builds on this evidence base regarding the need to explore factors affecting implementation and maintenance of TRT delivery across Sweden.

### 1.3. Models for Scale-Up of Interventions

Three main models for scaling up social programmes have been identified: (1) Branching pathways, (2) affiliate pathways, and (3) distribution network pathways (Figure 1) [19].

The branching pathway means that a lead organization expands and becomes able to offer the programme at multiple sites or to new target groups. In this type of pathway, the lead organization maintains responsibility for development, distribution, and implementation. This model minimizes variation, provides high level of control, but also slows down implementation.

The affiliate pathway refers to when potential implementers buy or license the rights from a lead organization to offer the social programme, including agreement to follow specific procedures and processes, not unlike the franchise concept in business. This allows for faster scale-up and less financial and other investment required from the original developer, retaining a certain degree of control, depending on how the contract, training, etc., are agreed on. However, the ultimate control for whether or not the programme is maintained is not with the lead organization, but the specific implementers.

A distribution network pathway is when the lead organization works with a distribution organization using the latter’s existing network of implementing organizations. Often the distribution partner is a national organization with many local member agencies. Thus, the developer supplies and supports the further development of the social programme, while a distribution partner delivers it, and other local partners support the use of the programme in organizations that adopt and implement the programme. This model allows the least control for the lead organization over the programme, but offers possibility for quick spread. To protect programme integrity and promote fidelity, the leader and distribution organizations need to provide guidance for adaptations and promote the forming of facilitator networks [19].

The model used in Sweden for scaling up TRT could be classified as the distribution network pathway. This is well in line with the model used elsewhere by the Children and War Foundation, who explicitly developed the programme for effective spread in low-resource settings. The Children and War Foundation provided the training and Uppsala University, with the support of private funders, conducted the translation and first pilot testing of the programme. Once it became clear that TRT was available for spread, we contacted Children’s Rights in Society (BRIS), a Swedish non-governmental organization (NGO). They then partnered with the Children and War Foundation to become the providers of training, responsible for spread and organizers of the national group leader network. They, in turn, worked with local organizations, such as municipalities, county councils, private providers, and other NGOs to implement the programme.

This study set out to evaluate the factors and agents that have facilitated the implementation and maintenance of TRT, a community-based intervention for refugee youth reporting symptoms of post-traumatic stress. Although over 300 practitioners have been trained in TRT in Sweden, successful implementation and maintenance of TRT has only been observed at a limited number of sites. The purpose was to shed light on the factors that have made this possible, using the idea of positive deviance. Successful sites were defined as: having conducted at least two TRT groups and maintaining full delivery of the programme. As the study was explorative, there were no hypotheses.

## 2. Materials and Methods

### 2.1. Participants

The inclusion criteria for sites were: (i) having conducted two or more TRT groups and (ii) planning to continue conducting groups in the future. The reason for these criteria was the experience that many sites had conducted one or two groups, but then discontinued hosting groups—hence, continuing after two groups was considered ‘successful’ implementation and maintenance. A survey with trained group leaders was conducted in December 2018. The group leaders were approached via the TRT network managed by BRIS, the partner NGO in the project. The survey yielded 16 sites with ongoing or previous TRT groups for refugee children and youth. Of these, six sites fulfilled the inclusion criteria and were included in the study. Two additional sites were initially interviewed but later had to be omitted, as they did not turn out to fulfil criteria; one site had not conducted two groups but was just recruiting and one had discontinued delivery after three successful groups.

A contact person at each of the identified sites was approached by email or telephone, with a request to interview one or several people with the experience of implementing and/or maintaining TRT groups at their site, and all agreed to participate. Seven participants from six sites in Sweden were interviewed for the study. Most of the participants were TRT group leaders themselves, except for one participant who was a manager. All participants were women. Five were counsellors with a social work background and two were registered nurses (Table 2). All participants gave their informed consent for inclusion before they participated in the study and a consent form was sent to the participants by email, to sign and return by mail, to conform with personal data guidelines of the General Data Protection Regulation. The study does not fall under the domain of the Swedish Law on Ethics including human subjects (2003:460) as no sensitive personal information was handled and practitioners were interviewed in their professional roles.

### 2.2. Data Collection

The seven semi-structured interviews were conducted by telephone and audio-recorded, with the consent of the participant. The interviews were conducted in Swedish, lasted between 30 min and 1 hour and were ended when the participant had nothing more to share. The audio recorded files and the transcribed documents were saved behind password protection by the first author (E.L.).

An interview guide which was constructed by the authors was used (Table 3). The guide ascertained that the included topics were discussed, but the participants were encouraged to talk freely about their work with TRT, what had worked well, and which factors had contributed to that. The questions in the interview guide were guided by the REP framework [3], in that every stage of the implementation process was considered.

### 2.3. Method of Analysis

The data were analysed using inductive content analysis as described by Graneheim and Lundman [20]. The interviews were listened to with the research questions in mind, were transcribed and the transcripts read several times. Translations into English were made for analysis, discussion, and publication. Meaning units were identified, condensed, and labelled with a code. The codes were then compared, with the whole text in mind, and sorted in categories. The initial manifest content analysis was conducted by the first author (E.L.), followed by an iterative process together with the last author (G.W.) to identify the latent theme and categories, verified and modified by the second author (A.S.) until consensus was reached.

## 3. Results

The analysis resulted in one overarching theme and three categories (Figure 2).

### 3.1. Theme: Active Networking and Collaboration

A reoccurring theme when investigating the factors of the sites that were successfully delivering TRT groups was the need for collaboration on multiple levels. The importance of networking and collaboration was not always brought up by the interviewed professionals themselves in response to the questions about which factors had been important for their delivery of TRT but was often implicit and identified as a latent theme during the analysis. For example, several of the sites had attempted different strategies in planning for and delivering the programme during their work with TRT. They expressed, when reasoning about the results of the different strategies, a better experience regarding most aspects when collaborating with other actors working with or for refugee youth as well as an intent to continue or increase their collaboration with others for future TRT groups.

All interviewed professionals described some collaboration with other organizations in their community. These collaborations were sometimes one-way information about the TRT groups, as a part of recruitment, but many collaborations were in different ways crucial for the groups to happen. One interviewed professional described that she and her colleague had received help in recruitment and finding facilities for the group, as well as partnering up with an organization who followed up on the children at the asylum accommodation centres. This was all achieved through their connections with civil society, church, and government organizations. She also described the general collaboration and communication between all kinds of organizations in the region as unusually good, which made further collaborations and common problem solving easier.

*“I think it is important that you cannot think strictly from a health care perspective or a civil society perspective (…) We have a well-established network that we really need to use in these situations”* (Part. 3, counsellor at health care centre).

### 3.2. Category: Going to Where the Potential Recipients Are

The interviewed professionals described access to TRT recipients and recruitment in very different words. Some found it very challenging and put emphasis on their specific solutions for recruitment as a success factor. Some described it as easy, when asked, but did not bring it up by themselves. The main difference between these groups was whether or not they attempted to access recipients, recruit them, and conduct groups where the recipients already were. For all sites where the groups were hosted in schools, asylum accommodation centres, or, to some extent, health clinics for asylum seekers and refugees, recruitment was not considered a challenge. An interviewed professional from a health clinic described their geographical area as having a high concentration of migrants and the youth were identified as potential TRT recipients during health screening or when seeking health care for stress-related symptoms.

*“We work at a unit exclusively for asylum seekers and our town is a town which receives a lot of asylum seekers, so I suppose we found it easy to recruit this entire time.”* (Part. 1, registered nurse at health care centre).

At another site, the TRT leaders had identified the need to offer the groups where the potential TRT recipients already were and had started to travel to asylum accommodation centres to conduct groups. Through collaborations with local organizations, facilities were arranged for. The interviewed nurse found it a very successful method as it minimized the practical barriers for the participants to join each group session. She also perceived the need for TRT groups at the accommodation centres to be great.

*“It’s actually a bit funny, for the last two groups we have screened and invited 7 or 8 people and we have ended up with 13 or 14 participants. They’ve multiplied [laughs].”* (Part. 3, counsellor at health care centre).

The interviewed professionals from the sites where TRT leaders did not meet potential TRT recipients in their ordinary work, described numerous, more or less successful, attempts to recruit and often having to re-evaluate their recruitment strategies. One of the sites had established a well-functioning collaboration with a school, starting with an agreement with key actors at the municipality level and thereby ensuring support on all levels down to the teachers at the school. All these sites described working hard for creating trusting relationships with potential recipients as well as identifying and engaging committed adults within the youths’ network, and that these collaborations were key aspects of recruitment to the groups.

*“What worked for us was a collaboration with the school, around the youth attending that school, and where we had a real enthusiast at the school, who helped us with recruitment of new youth, who was really selling it.”* (Part. 6, manager, unit for housing for refugee youth).

The same sites all had reached the conclusion that they would attempt to conduct TRT at a site where the potential recipients already are, through new collaborations, mostly with schools. An interviewed professional from an NGO that had attempted delivering TRT in several settings stated that the NGO had been important for the implementation and spread of TRT but now needed to hand it over, for a more stable, long-term delivery of the groups.

*“I believe that those who have their children in their everyday work should conduct the groups. If not, you are too far from them and you have to work really hard, harder than you should have to maybe.”* (Part. 2, counsellor, NGO collaborating with municipalities).

### 3.3. Category: Careful Integration of the Interpreter

Interpretation to another language was needed for almost all of the groups conducted at the sites. Thorough considerations were made around interpretation at all sites, to make the situations as good as possible for the TRT recipients and for the interpreters themselves as well as for the delivery of the programme. Although, the actual routines around interpretation varied between the sites.

At all the sites, the interviewed professionals made efforts to include interpreters whom they had worked with before and found suitable for working with sensitive topics. The exception was a rural site, where all individuals in a language group would know each other, who therefore worked exclusively with telephone interpreters. In the cases where TRT leaders were not familiar with the sites, they collaborated with a local partner whom they asked for advice. Interpreters were considered important for the TRT leaders not only for their role in translating what was said in a correct way, but for conveying the content in a culturally appropriate way, or “explaining rather than interpreting” as one participant phrased it. The participants describe the interpreter as a co-facilitator of the group rather than a language translation service, with a vital role in bridging the cultural gap between Swedish-born TRT leaders and migrant children, facilitating both differing mental health understandings, cultural pre-understandings and terminology.

Almost all interviewed professionals took actions to prepare the interpreter on the purpose and content of the group, but to what extent varied a lot. Some only presented the content briefly. Others sent written material to the interpreters beforehand and then spent several hours going through it together. It was considered important for the quality of the delivery of the programme, regarding what was actually conveyed to the group participants from the content of the manual when the group leader could not understand what the TRT recipients heard. Several interviewed professionals also pointed out that the interpreters needed to be aware of and comfortable with the content in the group since they might be suffering from traumatic memories themselves.

*“They have received the manual and read it, and then we sat down between an hour and several hours and went through it, with the interpreters. They really react to this considering their background, and it’s so important that they are stable and prepared for what is to come and that they can handle it.”* (Part. 3, counsellor at health care centre).

One interviewed professional described difficulty in following and guiding the conversation in the room when the TRT recipients spoke another language than her. An experience with having a TRT-trained Arabic speaking colleague joining as an assistant in one group had left a strong impression on her. Her Arabic speaking colleague had the benefit of both knowing TRT and being able to pick up on signs that something was misunderstood or needed further explanation. This was an example from an information meeting for parents of TRT recipients:

*“Then she told me that these two parents think they are here because their daughter is changing to another school. So, having someone who could pick that up. I would not have caught that because they were just chatting about it.”* (Part. 4, counsellor, NGO collaborating with schools).

In one group, the school’s language aid from the school interpreted during the TRT sessions. Although they were not a trained interpreter, it was perceived as positive since it made a long-term collaboration possible. The language aid interpreted in several groups after each other, came to know the manual well and built a trusting relationship to the interviewed professional, which allowed more freedom and creativity in the translations. Additionally, the language aid was present at the school when the TRT leaders were not and therefore able to follow up on the children in-between the sessions.

### 3.4. Category: Resource Availability and Management for Maintenance

The interviewed professionals had different agreements with their employers regarding TRT, but they usually found the amount of time allocated to TRT to be sufficient. Although, several mentioned that the time and energy spent on recruitment was often underestimated, both regarding the hard work to recruit but also the time needed to talk to all potential TRT recipients to know whether they were eligible for the group.

All sites except one had received external funding at some point for working with TRT, often as a result of collaboration with other organizations. Some interviewed professionals stated that the external funds had been crucial to start working with TRT. However, several also expressed concern that they might not be able to continue that work since these projects were soon coming to an end.

One interviewed professional, experienced in collaborating with several partners, said that organizations need to increase their flexibility to work with TRT. As examples, a TRT group leader might need to work later some days and receive compensation and have colleagues share tasks to allow time for the group. She also pointed out a change in resources allocated to refugee care compared to the time of the refugee crisis in 2015.

*“You have to have in mind that a few years ago things were different, when extra resources were added to the municipalities, such as personnel, integration coordinator and extra personnel to school etc., that is being removed more and more, and that makes our work more difficult, with school health personnel and language aides etc., gone.”* (Part. 2, counsellor, NGO collaborating with municipalities).

When asked about the future of TRT, one interviewed professional stressed the need for resources in future sites, as well as the aforementioned closeness to potential recipients, for a long-term engagement in offering TRT groups for refugee children.

*“I think that this revolves around that we need to hand it over to someone and the question is, who should we hand it over to? Is it the school or primary health care, who has the interest and the resources? We need help figuring that out.”* (Part. 5, counsellor, NGO collaborating with schools).

## 4. Discussion

The number of refugee minors who arrived in Sweden during 2015, and the difficulties for the healthcare and welfare systems to address their mental health problems, led to a need to provide and spread interventions effectively. The scale-up model used for TRT in Sweden can be classified as the distribution network pathway. This model makes it easy to achieve fast spread, but it comes at the cost of minimising the control over local implementation of the programme by the distributing organization. This might have made it more difficult for group leaders both to gain support when needed and to support each other, as there is no central organization. Additionally, the policy landscape has shifted since the programme was first introduced in Sweden. The need was at first more apparent in social work, but later shifted to schools and to health clinics dedicated to refugees [21]. A consequence of this is that trained group leaders have delivered few groups in a site and new leaders have been trained in another site.

The distribution network pathway, with both its possibilities and challenges, is well-reflected in the overarching theme, “Active networking and collaboration”. Network support was absolutely key in the success of these sites, thus demonstrating the extension of the distribution network model (Figure 1). Inter-professional and inter-organizational collaboration is considered an essential component of effective healthcare; however, the rules of collaboration and teamwork are often difficult to put into practice. Atwal and Caldwell [22] describe 4 prevailing issues when collaborating across professions: communication; documentation; conflict on goals; and organizational aspects. The findings from the present study indicated a few collaboration strategies that have been helpful in the implementation and maintenance of TRT: having an assigned person within an organization to take responsibility for the actions associated with the programme; being clear on the roles and responsibilities of each actor involved; and applying a flexible mindset to the process which involves a shared vision beyond professional or systemic differences. All of these strategies give rise to clear communication pathways and alignment of goals, which are central to effective collaboration [22].

The category “Careful integration of the interpreter” could be viewed as lack of support from a lead organization in developing the programme. The consideration required around interpretation is threefold: the interpreter should not come from the same community as the intervention recipients in order to allow recipients the freedom to share in a neutral space; the interpreter themselves should be comfortable with the content of the programme and have the opportunity to resolve any related personal issues or choose not to be involved if the content is too sensitive; the interpreter should be sensitive to the intended messages of the programme. Additionally, the interpreter was found to have a role as cultural broker, bridging a cultural gap between TRT group leaders and children. This is particularly interesting, as TRT leaders in an implicit way describe the interpreters as essential for the programme to be delivered. When working with an interpreter they had an established relation to, they could leave parts of programme delivery to them, which is closer to the role of co-facilitator than interpreter. This points out the need to include professionals with cross-cultural knowledge in development and delivery, to bridge cultural gaps and address potential differences in conceptualizations.

Interpretation can be challenging, particularly when psychological language is involved. Direct linguistic translations can sometimes result in the loss of intended meaning. In order to replicate the manualized TRT intervention, and to avoid unintended adaptations not in line with the programme theory, careful consideration should be given to interpretation. The findings from the present study indicated a successful strategy is to include interpreters in the preparation for TRT. Ideally, the interpreters will have an opportunity to be guided through the content of the programme thoroughly. This is something that could potentially be manualized and become part of the delivery of TRT for refugee populations. Additionally, training TRT leaders with the same cultural and language background as TRT recipients is a promising path. However, this needs to be balanced with the need to deliver TRT where the potential recipients are, for example through the personnel at their school.

The challenges conveyed in the present study related to the lack of a lead organization supplying necessary funds and support for maintenance. The additional resources allocated to Swedish municipalities and schools in response to the influx of refugee youth in 2015 enabled TRT delivery as an adjunct provision. Although many health and social care providers are of course affected by limited resources, these findings show that TRT, even though it is directed towards a group with major support needs and might prevent more serious mental health problems, are often funded with time-limited resources which made long term planning difficult. With the additional resources coming to an end, the case will need to be made for the continued delivery of TRT, which brings into light the ongoing issue of efficient use of limited resources in publicly funded healthcare systems. This would certainly align with the theme of “Resource availability and management for maintenance”, in which it was inferred that TRT, although a relatively “light-touch” intervention, requires greater resource than many anticipate, particularly relating to personnel time, and effective resource management is key to successful implementation and maintenance. The cost of continued delivery needs to be balanced with the potential consequences that could occur if the TRT model is not maintained.

A potential solution to some identified challenges can be found in the delivery of TRT groups online. This change in delivery format could help to alleviate challenges in each of the three categories. Firstly, it would make it easier for recipients to join the group from wherever they are, meaning that group leaders would not have to travel to recipients. This would both save resources and make recruitment for groups easier. Additionally, single-language groups could more easily be formed. However, it is reasonable to believe that networking and collaboration on multiple levels would still be essential for group leaders to access potential recipients.

The relatively small sample size is acknowledged; however, all potential sites that fulfilled inclusion criteria were included, and thus a total sample has been achieved. The present study does not tackle the earlier implementation considerations around intervention adoption as all the included sites have articulated a desire to deliver TRT, which implies a belief in the programme that may not be evident elsewhere in new sites. The purposive sampling procedure helped improve transferability; results are applicable in settings that have overcome certain challenges and manage maintenance, albeit not without difficulty, of TRT programme delivery. The first author (E.L), who conducted the interviews, is a trained TRT group leader. This might have led the interviewed professionals to wanting to appear successful, but it might also have added value as the interviewed professionals shared their experiences with someone who has conducted TRT groups and thus could ask more insightful follow-up questions.

This study only interviews TRT providers from successful sites, which is suitable for its aim and for implementations purposes. However, only one manager was included. On reflection, purposeful sampling of managers as well as TRT group leaders may have given greater insight on the strategic aspects of intervention maintenance. Another potential future direction for the research could be an assessment of the “unsuccessful” sites to explore whether the presence or absence of the identified factors, verifying whether or not they are key to successful maintenance of TRT delivery.

Additionally, to fully understand the usability and appropriateness of the programme, the recipients’ perspective is essential. A qualitative evaluation of the intervention was done, by the authors’ research group, with unaccompanied refugee youth in the exploratory trial [13], with promising findings. Continuing with qualitative studies with other potential recipients of the intervention, such as younger children in families, would be a valuable path for future research, including exploring the perceived cultural appropriateness of the intervention. Additionally, involving representatives from the refugee community in programme development and delivery as well as research, is just as important. In the ongoing trials of TRT [17,18], in which the authors are involved, users are represented by refugee parents and youth, who are involved in all major decision in the trials [23]. Through user involvement, the potential recipients’ voices are represented throughout the process and can potentially lead to more culturally appropriate programmes as well as more relevant and reliable research.

## 5. Conclusions

Our results indicated that active networking and collaboration was key to successful maintenance of TRT delivery, well in line with the distribution network pathway model employed for scale-up. Additional categories included “Going to where the potential recipients are” and “Resource availability and management for maintenance”, the former relating to the importance of networks and the latter to the challenges encountered due to lack of a lead organization supplying necessary funds and support for maintenance. The category “Careful integration of the interpreter” underlined the importance of interpreters, essentially as co-facilitators of the intervention. Although the interviewed professionals represented successful sites, they remained dependent on informal networks and collaboration for programme delivery. Online delivery of TRT is a potential solution to some of the identified challenges, an approach which should be developed together with representatives from the refugee community.

## Figures and Tables

**Figure 1 ijerph-18-00043-f001:**
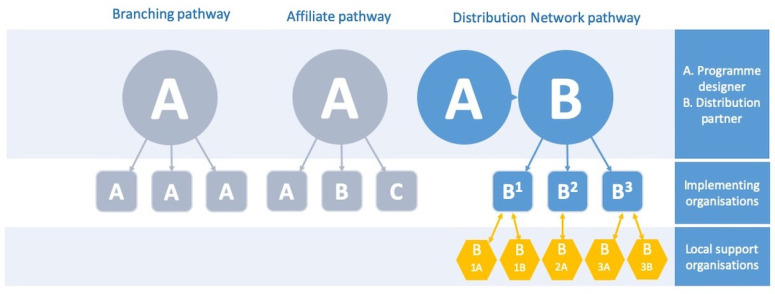
Branching, Affiliate, and Distribution Network Pathway Structures with an extension to local support organizations, as informed by the present study.

**Figure 2 ijerph-18-00043-f002:**
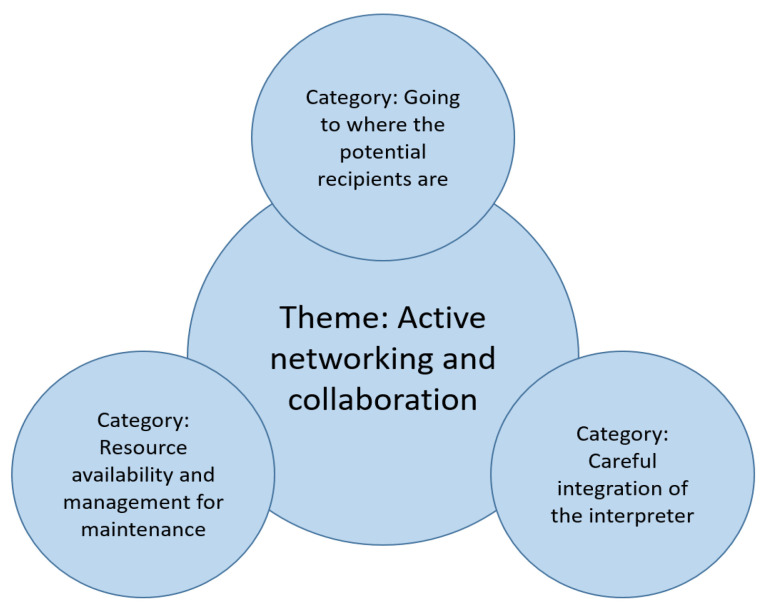
Overview of results.

**Table 1 ijerph-18-00043-t001:** Content of the Teaching recovery techniques (TRT) programme: the 5 child sessions.

Session and Theme	Techniques
Session 1. Intrusion	Psychoeducation: trauma events and reactions Normalization of traumatic stress reactions Establishing a safe place
Session 2. Intrusion	Imaginary techniques Dual attention tasks (EMDR-inspired) Dream work Time for bothering thoughts and worries
Session 3. Arousal	Psychoeducation: Arousal Muscle relaxation Positive self-talk The “fear thermometer” Sleep and activity planning
Session 4. Avoidance	Psychoeducation: Reminders and avoidance Mapping own reminders Introducing graded exposure
Session 5. Avoidance	Exposure to traumatic memories (drawing, writing, talking) Look to the future

**Table 2 ijerph-18-00043-t002:** The sites included in the study.

Setting	TRT Group Leader at the Site	Delivery	Participant Number
Health care clinic for asylum seekers	Counsellor and registered nurse	Clinic	Part. 1 (nurse)
Collaboration between an NGO and municipalities	Counsellors and a registered nurse	Housing units, facilities, and library	Part. 2 (counsellor)
Health care clinic for asylum seekers and refugees	Counsellors with therapeutic training	Clinic, asylum accommodation centre	Part. 3 (counsellor)
Collaboration between an NGO and a school	Counsellors	School	Part. 4 (counsellor) Part. 5. (counsellor)
Unit within the municipality responsible for housing for refugee youth (family homes, group homes)	Social workers and professionals with degrees in behavioural science and public health	Housing units, school	Part. 6 (manager)
Health care clinic for asylum seekers and refugees	Registered nurses	Clinic and municipality facilities	Part. 7 (nurse)

**Table 3 ijerph-18-00043-t003:** Interview guide.

Interview Guide	
Organization	Which type of organization do you work in? How was TRT introduced in your organization? How do you work with TRT today? Was there a process of buy-in around TRT in your organization?
Financing	How are your TRT groups financed? Which resources are allocated to TRT?
Recruitment	How did you recruit participants to your groups? What has been successful in recruitment and what has not? Did you collaborate with other organizations?
Including TRT in a stepped care model	What are your thoughts on the possibilities of including TRT in a stepped care model in your organization?
Other	Did you use interpreters in your groups? Are there any specific factors you think were important for making TRT work in your organization?
TRT in the future	What do you plan regarding future TRT groups? What do you need to continue working with TRT? Is there anything you want to add?

## Data Availability

The data generated during the current study are not publicly available due to the small sample size and risk for identification of participants, but are available from the corresponding author on reasonable request. The data is not publicly available due to ethical restrictions. The small data set contains information that could compromise the privacy of research participants. Requests to access the data should be directed to the corresponding author.

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
