# Peer review of "Implementation and Maintenance of a Community-Based Intervention for Refugee Youth Reporting Symptoms of Post-Traumatic Stress: Lessons from Successful Sites"

_ijerph, 2020, doi:10.3390/ijerph18010043_

Round 1

Reviewer 1 Report

Manuscript ID: ijerph-1005193

Title: Implementation and maintenance of a community-based intervention for refugee youth reporting symptoms of post-traumatic stress: Lessons from successful sites

Journal: International Journal of Environmental Research and Public Health (IJERPH)

I thank the IJERPH for the invitation to review the manuscript “Implementation and maintenance of a community-based intervention for refugee youth reporting symptoms of post-traumatic stress: Lessons from successful sites”. This manuscript addresses a relevant topic, which is of interest at an international level. Although the study was carried out two years ago, research related to refugees remains essential for the development and implementation of different forms of support and solidarity for people affected by conflict/war. The fact that this study was carried out in Sweden, a country with a considerable increase in asylum seekers, is another valuable aspect. However, a set of flaws compromises the quality of the manuscript.

Below are some noteworthy problems:

Abstract:

  1. The number of interviews conducted in this study must be reported.
  2. Instead of saying “Interview data were analysed using content analysis” (singular - it seems that only 1 interview was analyzed), it should be included a statement explaining how content analysis served to this study (e.g., to identify the latent theme and categories...).
  3. The results and conclusions (the authors call it "discussion") are not clear. For example, the sentence "Active networking and collaboration were key to successful maintenance of community-based delivery" should be presented as a conclusion in this section.

In relation to points 2 and 3: the abstract must be in accordance with what is presented in the manuscript.

  1. According to the Instructions for Authors, “The abstract should be a single paragraph and should follow the style of structured abstracts, but without headings”.

Introduction

  1. The sentence “Symptoms of post-traumatic stress are well-known sequels of war and refuge..” is confusing, since refuge means “(a place that gives) protection or shelter from danger, trouble, unhappiness, etc.” (Cambridge Dictionary). I believe the authors meant that symptoms of post-traumatic stress are common, or at least well-known, among refugees, people who leave their countries because they are at risk of serious human rights violations and persecution.
  2. When explaining the promising impact of TRT on clinical outcomes, the authors mention the abbreviations of the assessment instruments that are not described in the manuscript, or even written in full (e.g., DSRS and CRIES-13). In fact, there is no need to mention the instruments when presenting the outcomes of other studies, in the section dedicated to the literature review, unless it is relevant to the present study.
  3. The authors subdivided the introduction, which can make the text more objective. However, the headings (1.1 ... 1.4) can be confusing to the reader. I suggest reorganizing this section, perhaps a simple change of headings may be sufficient. It is unnecessary to use the heading "Aims" when presenting the objectives of the study.
  4. The purpose of this study is presented three times (lines 53-55; 121-123; 162-165). The objectives should be presented at the end of the introduction and the research hypotheses must be added.
  5. When addressing their research, the authors use “study” and “paper”. In order to maintain consistency throughout the manuscript, I suggest using “study”.
  6. Regarding the sentence “we contacted BRIS, the national NGO Children's Rights in Society”: Although the reader can understand the meaning of BRIS and NGO is a well-known acronym, abbreviations and acronyms should be defined in parentheses the first time they appear in the manuscript. Suggestion: “we contacted Children's Right in Society (BRIS), a Swedish non-governmental organization (NGO)”.

Materials and Methods

  1. In order to make the eligibility criteria for sample selection clear, I suggested reorganizing the first section (2.1 Participants). Before presenting the “16 sites with ongoing or previous TRT groups...”, the eligibility criteria for the selection of the sample must be clearly presented, justifying the reasons why the authors defined such criteria. Then, the number of sites that met the criteria should be presented, following the specific reasons for some of them to be later excluded from the analysis.
  2. Please define GDPR, as indicated above (abbreviations and acronyms).
  3. Avoid presenting unclear data, such as “At least one person from each site was approached and all agreed to participate”. The last paragraph of this section “Seven participants from six sites in Sweden were interviewed for the study” can be included here, while statements about informed consent and ethical issues can be placed as the last paragraph of the section on participants.

Results

  1. It is necessary to clarify whether the statement “The importance of networking and collaboration [...] was often implicit and identified as a latent theme during the analysis” (lines 259-262) is directly related to the following sentences: “Several of the sites [...] They expressed, when reasoning about the results of the different strategies, a better experience regarding most aspects when collaborating with other actors working with...” (lines 263-267).
  2. I strongly suggest the use of quotation marks when presenting the interviewees' statements, if there is no standard defined by the journal (lines 277-279; 291-293; 300-302; 311-313; 319-321; 346-349; 356-358; 380-383; 387-389). Sometimes, the reader can only realize that it is a quotation when he/she sees the parentheses at the end of the interviewee's statement.
  3. It may be easier for the reader if numbers or letters are assigned to respondents (interviewees). For example, a nurse's statement is presented (line 291), so the next paragraph starts with “At another site...” and describes a nurse's perception. It probably refers to another nurse, but it can be confusing. In addition, in the sequence of “She also perceived the need for TRT groups in the accommodation centers to be great”, another statement is presented, now a testimony from a “Counsellor at health care center”. At this point, we wondered if this is the place that had two respondents.

Discussion

  1. Instead of saying “Interview data from the present study” (lines 409 and 433), which also seems to refer to only 1 interview, I suggest using “The findings of the present study..." or "According to the content analysis, we found that...”
  2. The use of the expression “several analysts” is not correct, since they have 3 researchers/analysts involved in this study/analysis. In addition, the statement about the credibility is not necessary, as the interview script and the authors as analysts are aspects that can be questioned.
  3. It should be relevant to mention, more clearly, that the authors participate in the studies referred to between lines 474 and 480.
  4. It is not necessary to create a separate section in the discussion (Methodological considerations). In fact, I strongly suggest eliminating it.

Conclusion

The conclusions section is clear and objective.

Finally, I suggest a careful review of the English terms they use, even in the interviewees' statements, in order to ensure that they are in accordance with what they had intended to express. As the authors highlighted “Direct linguistic translations can sometimes result in the loss of intended meaning” (page 11).

I hope these comments are a useful guide for authors to improve the manuscript.

Author Response

Abstract. The number of interviews conducted in this study must be reported.

Response: This has been amended according to the reviewer’s suggestion (p.1, line 16).

Abstract: Instead of saying “Interview data were analysed using content analysis” (singular - it seems that only 1 interview was analyzed), it should be included a statement explaining how content analysis served to this study (e.g., to identify the latent theme and categories...).

Response: This sentence has been rewritten (p.1., lines 18 -19).

Abstract: The results and conclusions (the authors call it "discussion") are not clear. For example, the sentence "Active networking and collaboration were key to successful maintenance of community-based delivery" should be presented as a conclusion in this section.

Response: The abstract has been amended to be clearer to the reader (p.1, lines 11-28).

Abstract: In relation to points 2 and 3: the abstract must be in accordance with what is presented in the manuscript.

According to the Instructions for Authors, “The abstract should be a single paragraph and should follow the style of structured abstracts, but without headings”.

Response: The headings have been removed from the abstract (p.1, lines 11-28).

Introduction: The sentence “Symptoms of post-traumatic stress are well-known sequels of war and refuge..” is confusing, since refuge means “(a place that gives) protection or shelter from danger, trouble, unhappiness, etc.” (Cambridge Dictionary). I believe the authors meant that symptoms of post-traumatic stress are common, or at least well-known, among refugees, people who leave their countries because they are at risk of serious human rights violations and persecution.

Response: The sentence has been amended to refer to “forced migration” rather than “refuge” (p.2, lines 68-69).

Introduction: When explaining the promising impact of TRT on clinical outcomes, the authors mention the abbreviations of the assessment instruments that are not described in the manuscript, or even written in full (e.g., DSRS and CRIES-13). In fact, there is no need to mention the instruments when presenting the outcomes of other studies, in the section dedicated to the literature review, unless it is relevant to the present study.

Response: The instrument acronyms have been removed from the manuscript (p.3, line 128-131). 

Introduction: The authors subdivided the introduction, which can make the text more objective. However, the headings (1.1 ... 1.4) can be confusing to the reader. I suggest reorganizing this section, perhaps a simple change of headings may be sufficient. It is unnecessary to use the heading "Aims" when presenting the objectives of the study

Response: The headings in the Introduction have been changed to make the manuscript easier to follow for the reader (lines 66, 78, 142), with the exception of ‘Aims’ which has been removed (line 185). 

Introduction: The purpose of this study is presented three times (lines 53-55; 121-123; 162-165). The objectives should be presented at the end of the introduction and the research hypotheses must be added.

Response: The first sentence has been removed (line 65), the second has been amended to not state the aim but still add to readability of the text (lines 139-141), and some of the text has been moved to the last  section of the Introduction (lines 185-197). As it was an explorative study, there were no hypotheses. This has been added to the manuscript (lines 196-197).

Introduction: When addressing their research, the authors use “study” and “paper”. In order to maintain consistency throughout the manuscript, I suggest using “study”.

Response: The manuscript has been thoroughly reviewed for inconsistencies, to make sure only the word “study” is used, as suggested.

Introduction: Regarding the sentence “we contacted BRIS, the national NGO Children's Rights in Society”: Although the reader can understand the meaning of BRIS and NGO is a well-known acronym, abbreviations and acronyms should be defined in parentheses the first time they appear in the manuscript. Suggestion: “we contacted Children's Right in Society (BRIS), a Swedish non-governmental organization (NGO)”.

Response: This sentence been amended according to the reviewer’s suggestion (p.4, line 179-181).

Materials and Methods: In order to make the eligibility criteria for sample selection clear, I suggested reorganizing the first section (2.1 Participants). Before presenting the “16 sites with ongoing or previous TRT groups...”, the eligibility criteria for the selection of the sample must be clearly presented, justifying the reasons why the authors defined such criteria. Then, the number of sites that met the criteria should be presented, following the specific reasons for some of them to be later excluded from the analysis.

Response: The ’Participants‘ section has been reordered and amended according to these suggestions (p.5, lines 201-221).  

Materials and Methods: Please define GDPR, as indicated above (abbreviations and acronyms).

Response: This sentence been amended according to the reviewer’s suggestion (p.5, line 219).

Materials and Methods: Avoid presenting unclear data, such as “At least one person from each site was approached and all agreed to participate”. The last paragraph of this section “Seven participants from six sites in Sweden were interviewed for the study” can be included here, while statements about informed consent and ethical issues can be placed as the last paragraph of the section on participants

Response: The ‘Participants’ section has been reordered and amended according to these suggestions (p.5, lines 201-221).   

Results: It is necessary to clarify whether the statement “The importance of networking and collaboration [...] was often implicit and identified as a latent theme during the analysis” (lines 259-262) is directly related to the following sentences: “Several of the sites [...] They expressed, when reasoning about the results of the different strategies, a better experience regarding most aspects when collaborating with other actors working with...” (lines 263-267).

Response: These sentences are directly related and the text has been rearranged accordingly (lines 417-419).

Results: I strongly suggest the use of quotation marks when presenting the interviewees' statements, if there is no standard defined by the journal (lines 277-279; 291-293; 300-302; 311-313; 319-321; 346-349; 356-358; 380-383; 387-389). Sometimes, the reader can only realize that it is a quotation when he/she sees the parentheses at the end of the interviewee's statement.

Response: We have added quotation marks to the quotes to improve readability, as suggested (lines 435, 450, 461, 477, 481, 487, 515, 530, 557, 567).

Results: It may be easier for the reader if numbers or letters are assigned to respondents (interviewees). For example, a nurse's statement is presented (line 291), so the next paragraph starts with “At another site...” and describes a nurse's perception. It probably refers to another nurse, but it can be confusing. In addition, in the sequence of “She also perceived the need for TRT groups in the accommodation centers to be great”, another statement is presented, now a testimony from a “Counsellor at health care center”. At this point, we wondered if this is the place that had two respondents.

Response: We agree that this will improve the readability of the text. We have assigned numbers to each of the participants and added these to Table 2 (line 360) and to the quotes  (lines 435, 450, 461, 477, 481, 487, 515, 530, 557, 567).

Discussion: Instead of saying “Interview data from the present study” (lines 409 and 433), which also seems to refer to only 1 interview, I suggest using “The findings of the present study..." or "According to the content analysis, we found that...

Response: We have changed the mentioned sentences (p. 11, lines 592 and 616).

Discussion: The use of the expression “several analysts” is not correct, since they have 3 researchers/analysts involved in this study/analysis. In addition, the statement about the credibility is not necessary, as the interview script and the authors as analysts are aspects that can be questioned.

Response: The sentence has been removed from the manuscript (p.12, lines 655).

Discussion: It should be relevant to mention, more clearly, that the authors participate in the studies referred to between lines 474 and 480.

This is a relevant comment and we have amended the text to make this clear (p.12, lines 667 and 673).

Discussion: It is not necessary to create a separate section in the discussion (Methodological considerations). In fact, I strongly suggest eliminating it.

Response: The sudheadingMethodological considerations’ has been removed (p.12, line 649).

Conclusion: I suggest a careful review of the English terms they use, even in the interviewees' statements, in order to ensure that they are in accordance with what they had intended to express. As the authors highlighted “Direct linguistic translations can sometimes result in the loss of intended meaning” (page 11).

Response: We agree that translations between languages can be difficult. We have reviewed the English terms used and feel comfortable they align with what was intended.

Reviewer 2 Report

This is a well-written paper describing factors influencing the successful implementation of a community-based intervention for refugee youth in Sweden. The study is informative and will be useful in contexts that may be implementing similar programmes. The study limitations are clearly and thoughtfully identified with directions for future research described. I only have two comments/ suggestions: 

1) For readability, it would be helpful to more clearly distinguish the quotations from the main body of the text by using full intent, line breaks, and/ or quotation marks in keeping with journal formatting standards. 

2) The discussion of the role of interpreters, their importance as co-facilitators and in understanding nuances of the discussion as described on page. 9, line 350 raises the question of whether training native language speakers in TRT should be prioritized. Given that the program is offered by non-specialists with a three-day training session, would the authors recommend prioritizing task-shifting to community providers who are fluent in the relevant languages? This seems like it might further improve the quality and acceptability of the TRT program. 

Author Response

For readability, it would be helpful to more clearly distinguish the quotations from the main body of the text by using full intent, line breaks, and/ or quotation marks in keeping with journal formatting standards.

Response: As mentioned above, we have added quotation marks to the quotes to improve readability (lines 435, 450, 461, 477, 481, 487, 515, 530, 557, 567).

The discussion of the role of interpreters, their importance as co-facilitators and in understanding nuances of the discussion as described on page. 9, line 350 raises the question of whether training native language speakers in TRT should be prioritized. Given that the program is offered by non-specialists with a three-day training session, would the authors recommend prioritizing task-shifting to community providers who are fluent in the relevant languages? This seems like it might further improve the quality and acceptability of the TRT program.

Response: This is a highly relevant point and we have added a short section about this (p.11, lines 620-622).

Reviewer 3 Report

I recommend publication of this article after revision: I would like a more critical analysis by the authors in the paragraph "Category: Resource availability and management for maintenance" and in their conclusions also to raise awareness to find and implement useful strategies for TRT deployment. In my opinion, the consequences that could occur if the TRT model is not maintained should also be briefly mentioned. 

Author Response

I like a more critical analysis by the authors in the paragraph "Category: Resource availability and management for maintenance" and in their conclusions also to raise awareness to find and implement useful strategies for TRT deployment. In my opinion, the consequences that could occur if the TRT model is not maintained should also be briefly mentioned.

Response: This category has been elaborated on in the Discussion section (p. 11, line 626-629). Although we feel exploration of the consequences that could occur if the TRT model is not maintained is beyond the scope of the current study, we agree that it is an important aspect to consider and we have now briefly mentioned it in the manuscript (p.12, lines 639-641).